# Effects of symbiotic population impairment on microbiome composition and longevity of *Bactrocera dorsalis*

Mazarin Akami[1,2,3], Ousman Tamgue[2], Xueming Ren[1], Yaohui Wang[1], Xuewei Qi[1], Koanga Mogtomo Martin Luther[2], Rosalie Annie Ngono Ngane[2] and Chang-Ying Niu[1]

[1]Hubei Key Laboratory of Insect Resource Application and Sustainable Pest Control, College of Plant Science and Technology, Huazhong Agricultural University, Wuhan 430070, People's Republic of China
[2]Department of Biochemistry, Faculty of Science, University of Douala, PO Box 24157, Douala, Cameroon
[3]Department of Plant Protection, Institute of Vegetables and Flowers, Chinese Academy of Agricultural Sciences, Beijing, People's Republic of China

MA, 0000-0003-2162-2119; C-YN, 0000-0003-2793-1422

**Subject Areas:**
microbiology/genomics

**Keywords:**
gut microbiome, aposymbiotic, axenic, longevity, *Bactrocera dorsalis*

**Author for correspondence:**
Chang-Ying Niu
e-mail: niuchangying88@163.com

In order to understand the role of symbionts for their insect hosts, it is customary to treat them with antibiotics or to sterilize eggs (treatments), resulting in aposymbiotic and axenic insects, respectively. Such axenic insects can then be compared with untreated controls. Fruit flies often bear complex communities which are greatly reduced by such treatments. However, the bacterial community is not completely eliminated. Here, we examine the effect of these procedures on the structure of the remaining bacterial communities in *Bactrocera dorsalis* (Diptera: Tephritidae) and on the insect longevity. The antibiotics (Norfloxacin and Ceftazedime) were administered to 1-day-old adult flies through sugar meal for 7 days, and eggs were surface sterilized and dechorionated to produce axenic lines. The flies were starved of protein before they were offered full diets or diets containing non-essential amino acids only. Antibiotic and egg disinfection treatments resulted in a significant reduction of the vast majority of gut bacterial populations, especially Proteobacteria, Firmicutes and Bacteroidetes. On the other hand, antibiotic allowed the persistence of Actinobacteria, Cyanobacteria and Acidobacteria populations. In untreated control flies, longevity was extended

irrespective of diet quality in comparison to treated flies. Conversely, when gut bacteria were largely reduced (aposymbiotic and axenic flies), longevity was reduced in the non-essential amino acids diet treatment versus slightly improved in the presence of a protein diet. We discuss these results in an ecological–life-history perspective.

# 1. Introduction

Insect guts can harbour highly diverse microbial communities that deeply influence their fitness parameters in a variety of ways [1–8]. Fruit flies belonging to the Tephritid family (Insecta: Diptera) harbour a variety of gut bacteria [9–11], which contribute to their nutritional ecology [3,12,13]. These symbionts have been shown to complement essential amino acids which are marginal or absent in the host diet, with the effect of enhancing protein synthesis [12], female fecundity [3] and host longevity [14,15].

Gut bacteria are generally identified and characterized by either culture dependent or molecular techniques, both targeting the 16S rRNA genes. But, the advent of high-throughput sequencing provides the ability to uncover a much larger fraction of the microbial diversity associated with arthropods, enabling the identification of novel species [11,16–18].

A growing number of studies have indicated that bacterial elimination by using antibiotics or by eggs dechorionation affects behavioural and physiological parameters of many fruit flies. For example, depending on the nutritional status, the suppression of gut bacteria reduced female fecundity of *Bactrocera oleae* [12] and *Bactrocera minax* [14] and greatly affected the reproduction capacities of *Drosophila melanogaster* [8,19] even when the adults were fed with full diets. The abundance of microbial populations in *Ceratitis capitata* and *D. melanogaster* highly impacted their lifespan in a diet-dependent fashion [1,15,20]. The administration of antibiotics oxytetracycline and sulfanilamide in adult diet reduced the midgut microbiota (abundance and diversity) and larval survival of the melon fly *B. cucurbitae* [21]. Similarly, the antibiotic oxytetracycline reduced the gut microbiome and survival of the wax moth *Galleria mellonella* in a dose-dependent manner, and deterred the expressions of tetracycline-resistance genes [22]. Abundant reports highlighted the effects of antibiotics on microbiota of flies and other insects (cabbage root fly, *Delia radicum* [23], melon fly, *D. melanogaster* [24] and diamondback moth *Plutella xylostella* [25]).

Axenic flies are produced from sterilized eggs (using bleach and ethanol) which develop on autoclaved diet and are maintained under aseptic conditions [20,26]. Bleaching removes the embryonic chorion, suppressing egg-coating bacteria except those vertically transmitted such as *Wolbachia* and *Spiroplasma* [27], resulting in bacteria-free insect lines. However, the bacterial community is not eliminated completely and residual populations usually remain [12,15,28]. Although few studies investigated the composition, density and diversity of gut microbial communities of antibiotics-fed insects [22,28,29], the magnitude and direction of changes remain unknown in most cases. The objective of our study was to outline if the two ways of reducing gut bacteria produce predictable differences in bacterial composition. To explore the effect of such community crash (i.e. rapid and drastic reduction in symbiont community size), we measured fly longevity. As the microbiome affects the fly's nutrition, we further explored how diet quality affects the lifespan of *Bactrocera dorsalis* suffering drastic bacterial community reduction versus controls (symbiotic ones).

More specifically, we characterize gut bacteria of antibiotics-treated and egg-dechorionated *B. dorsalis* by using high-throughput 16S rRNA gene sequencing and predict that (i) the antibiotics and egg dechorionation produce similar bacterial structure and composition, (ii) divergence in bacterial communities will be observed in relation to the symbiotic flies, and finally (iii) gut bacterial disruption will affect the host longevity in a diet-dependent fashion. To verify this, we first compared the gut microbiome from three treatment groups (symbiotic, aposymbiotic and axenic), before comparing the gut microbiome of aposymbiotic and axenic flies, and second, we evaluate the treatment and diet effects on fly longevity. The overall idea withstanding our study is to screen the existence of bacterial species resistant to antibiotics and/or egg dechorionation, to evaluate the efficacy of these two sterilization methods commonly used in studying the host fitness bacterially mediated, and finally to see whether the presence or absence of some bacterial clades could be ecologically relevant in predicting the fly longevity.

# 2. Material and methods

## 2.1. Fly rearing and maintenance

*Bactrocera dorsalis* wild larvae were collected from infested orange fruits in the experimental orchard of Huazhong Agricultural University (30°4′ N, 114°3′ E) (Wuhan, Hubei Province, China) in September 2014 and were reared as previously described [16]. Briefly, third instar larvae were allowed to exit the fruits, pupate and eclose in sterile sands under controlled laboratory conditions (12 : 12 light–dark photoperiod; temperature $25 \pm 3°C$, and $67 \pm 5\%$ relative humidity). The resulting adults were maintained under an artificial diet as previously described [30]. The diet and water were provisioned (ad libitum, 1 l day$^{-1}$ each) through cotton wool.

## 2.2. Treatment diets

Three experimental diets were made (a full diet, only non-essential amino acids diet and a sugary diet) according to protocols described in previous studies [30,31]. All experimental flies (symbiotic, aposymbiotic and axenic) were fed a sugar diet before being offered either a full diet, made up of complete amino acids (essential and non-essential ones), minerals, salts and sugar in defined proportions, or a diet made up of only non-essential amino acids, minerals, salts and sugar in defined proportions (see details in electronic supplementary material, table S2).

## 2.3. Production of symbiotic and aposymbiotic flies

Symbiotic and aposymbiotic flies were produced from the wild larvae as mentioned above. Newly emerged 1-day-old flies were sex-separated and divided into two groups of 30 flies each (symbiotic and aposymbiotic). Flies were grown in 9 cm Petri dishes and fed with a sugar diet for 7 days or with sugar meal amended with antibiotic solutions (3 µg ml$^{-1}$ Norfloxacin and 5 µg ml$^{-1}$ Ceftazedime, selected after a preliminary susceptibility test on some bacterial strains [30]) for 4 days (starting from the first eclosion day), for the symbiotic control and for the aposymbiotic group, respectively. From the 5th to the 7th day, the antibiotics-treated group were shifted to antibiotic-free sugar diet to minimize the effects of antibiotics on host fly and the appearance and proliferation of antibiotic-resistant bacterial strains [1] (with modifications).

## 2.4. Production of axenic flies

Newly laid eggs (2 h old) were collected, serially disinfected and dechorionated as described elsewhere [31]. The resulting adults were maintained under sterile sugar diets for 7 days prior to bioassays and high-throughput community analyses. Egg sterility was checked and obtained (electronic supplementary material, S1).

## 2.5. Measurement of gut bacterial community size

Community size was measured using qPCR. Five adult flies from each treatment group (symbiotic, aposymbiotic and axenic) were collected; their guts were dissected and were individually and separately homogenized in Eppendorf tubes containing 500 µl PBS. Bacterial genomic DNA was extracted using the CTAB/SDS method [32], and the recovered DNA samples were checked for quality by measuring the absorbance at 260/280 and 260/230 nm using a spectrophotometer. qPCR was performed with 5 ng DNA per reaction, using SYBRGreen for tracking, in 384-well plates on an ABI 7300 Real-Time PCR detection system (Applied Biosystems Inc, Foster City, CA, USA). The 16S qPCR primers 1369_F (5′-CGGTGAATACG TTCYCGG-3′) and 1492_R (5′-GGWTACCTTGTTACGACTTN-3′) were used for measuring bacterial community size and the primers Bacdo_F (5′- TCTATATTCGCGCGGTTGCT-3′) and Bacdo_R (5′- GCGGGTGTGTTGAA GGTTTC-3′) of *Bactrocera dorsalis* actin gene (cytoplasmic A3a, XM_011200225.2) were used to normalize the bacterial content by constructing a standard for the 16S rRNA genes under the following cycling conditions: initial incubation at 95°C for 2 min, followed by 40 cycles at 95°C for 15 s and at 55°C for 1 min 30 s.

## 2.6. Longevity assay

The 7-day-old, sugar-fed adult flies were divided into two subgroups of 30 flies each. One group was shifted to the full diet, and the other to the non-essential amino acid diet. Longevity was measured by counting dead flies until population extinction in all treatments.

## 2.7. High-throughput community analyses of gut samples

The gut bacterial community structure of protein-starved flies, i.e. fed a sugar diet for 7 days (as described above) was analysed. Each treatment group (symbiotic, aposymbiotic and axenic) comprised three biological replicates of five pooled flies each. Gut homogenates from each biological replicate and treatment were used for DNA extraction and sequencing.

## 2.8. Genomic DNA extraction and amplicon generation

Total genome DNA from samples was extracted using the CTAB/SDS method [32]. DNA quality was checked on 1% agarose gels using a ladder and the purity was checked as above. DNA was diluted to 1 ng µl$^{-1}$ with sterile distilled water.

The V1-V3 variable region of the bacterial 16S rDNA gene was amplified to construct a gene library using bar-coded and broadly conserved primers for the PCR reaction: 27F_5′ <u>CCTATCCCCTGTGTG CCTTGGCAGTCTCAG</u>AGAGTTTGATCCTGGCTCAG-3′, and 533R_5′-<u>CCATCTCATCCCTGCGTGTCT CCGACGACT</u>NNNNNNNNNTTACCGCGGCTGCTGCAC-3′ [16], containing the A and B sequencing adaptors (454 Life Sciences) to facilitate pooling, segregation, sequencing and amplification of approximately 536 bp region of the mentioned gene ('Ns' represent the 8 nt barcode sequence for multiple samples while the underlined sequences represent the A-adaptor). All PCR reactions were carried out with Phusion® High-Fidelity PCR Master Mix with GC Buffer (New England Biolabs, Ipswich, MA, USA) and high-fidelity polymerase (New England Biolabs).

The PCR conditions were as follows: initial denaturation at 94°C for 2 min, followed by 30 cycles of denaturation at 94°C for 1 min, annealing at 60°C for 30 s and extension at 72°C for 1 min, and a final extension step of 10 min at 72°C. PCRs of DNA-free samples were run to check the potential contamination of buffers and primers [33]. PCR amplicons were later subjected to electrophoresis on a 2% agarose gel, stained with ethidium bromide, and the targeted fragment size (400–450 bp) was extracted, purified with Qiagen Gel Extraction Kit (Qiagen, Germany) and quality checked prior to pyrosequencing. Amplicon libraries were generated using TruSeq® DNA PCR-Free Sample Preparation Kit (Illumina, USA) following the manufacturer's recommendations, and index codes were added. Library quality was assessed on the Qubit@ 2.0 Fluorometer (Thermo Scientific) and Agilent Bioanalyzer 2100 system using a DNA1000 lab chip (Agilent), respectively. The library was then amplified by emulsion PCR before 454 pyrosequencing was performed from the A-end on an Illumina HiSeq2500 platform using a GS FLX Titanium system according to the manufacturer's instructions (Roche 454 Life Sciences), and 250 bp paired-end reads were generated.

## 2.9. Bioinformatics analyses

Paired-end reads were assigned to samples based on their unique barcode and truncated by cutting off the barcode and primer sequence. Paired-end reads were merged using FLASH 1.2.7 [34]. Quality filtering on the raw tags was performed under specific filtering conditions to obtain the high-quality clean tags [35] using QIIME 1.7.0 [36] quality-controlled process [37]. The tags were compared with the reference database using UCHIME algorithm to detect and remove chimera sequences before obtaining effective tags [38].

Operational taxonomic units (OTUs) analyses were performed by UPARSE 7.0.1001 [39]. Sequences with greater than or equal to 97% similarity were assigned to the same OTUs. Representative sequence for each OTU was screened for further annotation, and the GreenGene Database [40] was used based on the Ribosomal Database Project classifier 2.2 algorithm [41] to annotate the taxonomic information. OTUs abundance information was normalized using a standard of sequence number corresponding to the sample with the least sequences. The Good's coverage, the abundance-based coverage estimator (ACE), the bias-corrected Chao1 richness estimator, the jackknife estimator of species richness and the Shannon–Weaver and Simpson diversity indices were calculated with the Mothur package (table 1).

## 2.10. Statistical analyses

All mortality data were tested for homogeneity of variances using Levene's tests (otherwise the mortality data were log-transformed). To determine the important factors that shape fly survival, variables of daily living flies were analysed using the Cox's regression model (SPSS 20.0 software) with sex, diet types and symbiotic status as effects. The one-way analysis of variance (ANOVA) was used to analyse differences in

**Table 1.** Alpha diversity indices table showing the diversity and species richness of *B. dorsalis* gut microbiota in symbiotic, aposymbiotic and axenic adult flies. Values within columns with different letters are statistically different after Tukey HSD test at $p < 0.05$; $\alpha$: average number of OTUs observed in each sample (alpha diversity).

| samples | observed OTUs$^{\alpha}$ | Shannon | Simpson | Chao1 | ACE | goods_coverage (%) |
|---|---|---|---|---|---|---|
| symbiotic | 970 ± 152a | 6.46 ± 0.31a | 0.95 ± 0.01a | 1090 ± 176a | 2101 ± 853a | 0.994 ± 0.00a |
| aposymbiotic | 452 ± 20b | 4.77 ± 0.51b | 0.65 ± 0.03b | 495 ± 14b | 502 ± 18b | 0.996 ± 0.001a |
| axenic | 312 ± 33b | 3.29 ± 0.69b | 0.58 ± 0.10b | 357 ± 42b | 343 ± 50b | 0.998 ± 0.00b |

mortality and sequencing data among treatments using SPSS 20.0 software (Statsoft Inc, USA), followed by Tukey's test (HSD) at $p = 0.05$ for multiple comparisons. The relatedness between bacterial communities from different samples was determined based on OTU richness or abundance (Bray–Curtis dissimilarity). The analysis of molecular variance (AMOVA), analysis of similarity (ANOSIM) and UniFrac analysis (unweighted and weighted) were performed to evaluate the effects of treatments on *B. dorsalis* gut bacterial community among samples using the software PRIMER 7.0. The generated *phylip*-formatted distance matrix (Bray–Curtis dissimilarity matrix) was used for principal coordinate analysis (PCoA) to determine microbial community differences among samples. Sequences showing greater than 1% of bacterial family abundance in each sample were assigned for non-metric multi-dimensional scaling using QIIME 1.7.0 software and displayed with R software 2.15.3. The Pearson chi-square test was used to evaluate the effects of antibiotics and dechorionization treatments on gut bacterial community structure and composition. All results (alpha and beta diversity, survival and sequence data) are presented as mean ± s.d. of the three biological replicates. OriginPro 8.5.1 was used to draw longevity graphs.

# 3. Results

## 3.1. Bacterial species richness and diversity

A total of 58 926 raw sequences were generated from each of the nine biological replicates assigned for 454-pyrosequencing analyses. After demultiplexing, quality filtering and chimera removal of the sequencing data, an average of 56 559 clean reads per sample were retained. Subsequent clustering at 3% distance resulted in 574 ± 73, 441 ± 56 and 229 ± 46 OTUs from symbiotic, aposymbiotic and axenic flies, respectively (figure 1) (average relative abundance ± s.d. between biological replicates). The average read length was 242.6 ± 12.37 bp and the minimal number of reads per OTU was approximately 100 bp. Aposymbiotic and axenic flies shared most of their OTUs (330 and 310 OTUs, respectively) while symbiotic flies possess the highest number of unique OTUs (1251 OTUs) (figure 1). The shared OTUs between all treatments were 47 (electronic supplementary material, table S1).

The alpha diversity analysis indices (Shannon, Simpson, Chao1, ACE and Goods_coverage) of different samples at the 97% consistency threshold are shown in table 1. The bacterial diversity indices (Chao1 and ACE) were higher in symbiotic flies (ANOVA, d.f. = 2; $F = 9.014$; $p = 0.016$) as compared with aposymbiotic and axenic flies combined, in which the diversity was similarly lower (Chi-square, $\chi^2 = 14.87$, $R^2 = 0.89$, $p = 0.083$; table 1). The non-parametric richness indexes (Shannon and Simpson) at 3% distance showed that symbiotic flies exhibited greatest bacterial OTUs richness (ANOVA, d.f. = 2; $F = 13.796$; $p = 0.01$) in comparison with the aposymbiotic and axenic flies combined (Chi-square, $\chi^2 = 11.39$, $R^2 = 0.987$, $p = 0.006$) (table 1). In summary, much lower bacterial diversity and species richness were recorded in aposymbiotic and axenic samples compared with symbiotic ones.

## 3.2. Community structure analyses

The multi-response permutation procedure analysis based on Bray–Curtis distances revealed no significant difference between the microbiomes of aposymbiotic and axenic flies ($A = 0.08102$; observed-delta = 0.7518; expected-delta = 0.7519; $p = 0.4$). However, the microbiome of symbiotic flies was shown to be significantly different from that of aposymbiotc and axenic flies, respectively (Bray–Curtis distance statistics, $A = 0.009052$; observed-delta = 0.7332; expected-delta = 0.8062; $p = 0.02$ and $A = 0.0001318$; observed-delta = 0.7837; expected-delta = 0.8527; $p < 0.01$). The results

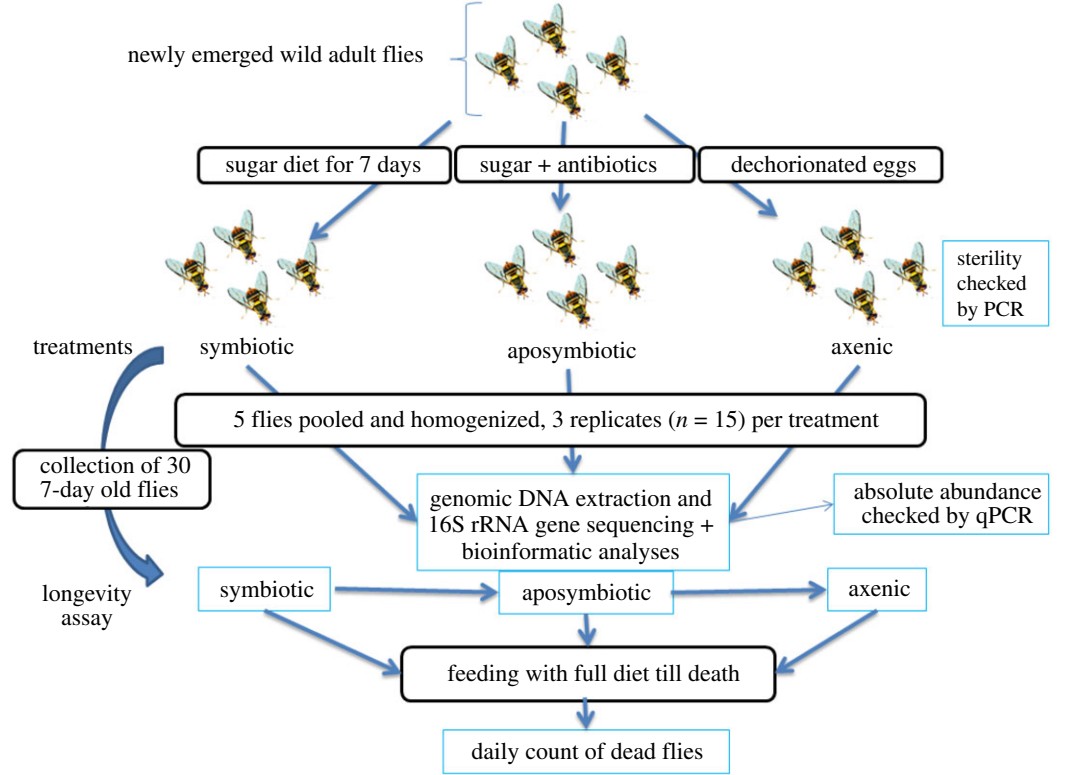

**Figure 1.** Flowchart of the experimental design.

indicated that there were significant differences between the bacterial community of symbiotic and aposymbiotic flies (S-Ap: $R = 0.7407$; $p = 0.006$) and between symbiotic and axenic B. dorsalis (S-Ax: $R = 0.978$; $p = 0.003$). But, no significant difference was recorded in the structure, composition and abundance of aposymbiotic and axenic flies ($R = 0.5185$, $p = 0.301$) (figure 2).

## 3.3. Species accumulation

According to rarefaction, coverage was $0.994 \pm 0.006\%$ in the symbiotic controls, $0.996 \pm 0.001\%$ in aposymbiotic flies and $0.998 \pm 0.006\%$ in axenic flies (table 1). This indicates almost complete coverage of 16S rRNA gene sequences (electronic supplementary material, figure S1a). The measure of OTUs number and the type of their distribution (species rank abundance curves) in symbiotic samples exhibit larger width, indicating higher species richness compared with aposymbiotic and axenic ones, respectively (electronic supplementary material, figure S1b).

## 3.4. Taxonomic affiliation

The bacterial communities in symbiotic flies were dominated by Proteobacteria ($70.43 \pm 6.2\%$), followed by Firmicutes ($13.45 \pm 1.25\%$) and Bacteroidetes ($8.70 \pm 0.98\%$). Other bacterial phyla consisted of Actinobacteria ($4.43 \pm 1.36\%$), Cyanobacteria ($1.65 \pm 0.55\%$), Acidobacteria ($1.13 \pm 0.2$) and Planctomycetes, Verrucomicrobia, Fusobacteria and Chloroflexi (all combined accounted for $1.44 \pm 0.15\%$ of the total abundance), and unclassified taxa (others) accounting for $3.84 \pm 1.27\%$ (figure 3). However, Proteobacteria, Firmicutes and Bacteroidetes phyla (which were dominantly represented in symbiotic flies) were significantly reduced in aposymbiotic and axenic flies (ANOVA, d.f. = 8; $F = 263.188$; $p < 0.0001$ and d.f. = 8; $F = 263.188$; $p < 0.0001$, respectively), but their proportions in aposymbiotic and axenic flies were not statistically different (Chi-square, $\chi^2 = 11.12$, $R^2 = 0.985$, $p = 0.053$). The phyla Actinobacteria ($64.77 \pm 3.99\%$ and $66.34 \pm 1.45\%$), Cyanobacteria ($24.97 \pm 3.92\%$ and $31.72 \pm 2.48\%$) and Acidobacteria ($11.03 \pm 1.84\%$ and $3.55 \pm 0.91\%$), accounted for the vast majority of sequence reads (greater than 70%) in aposymbiotic and axenic flies, as their relative abundance significantly increased compared with symbiotic ones (ANOVA, d.f. = 2; $F = 4.967$; $p < 0.0001$ and d.f. = 2; $F = 10.204$; $p < 0.0001$, respectively) (figure 3). In summary, the three most abundant bacterial phyla in the control symbiotic flies (Proteobacteria, Firmicutes and

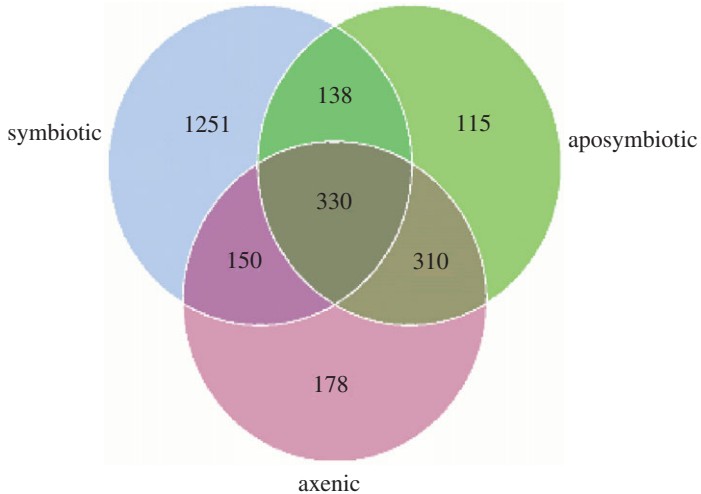

**Figure 2.** Venn diagram of OTU distribution of treatment-specific and shared OTUs.

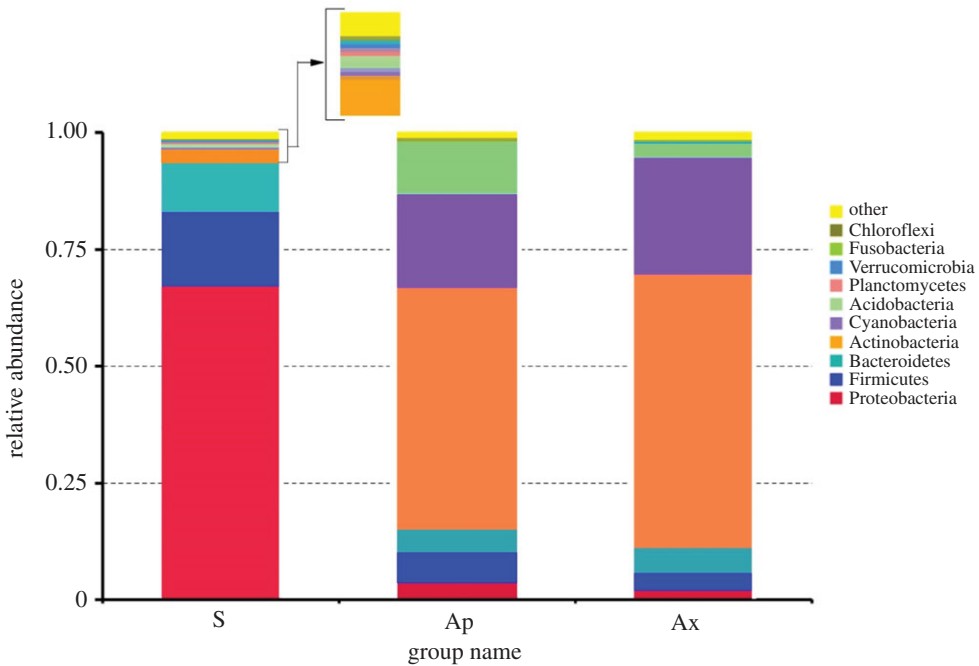

**Figure 3.** Bacterial community composition at the phylum level as revealed by 454-pyrosequencing. Phylogenetic groups accounting for less than 1% of all classified sequences are summarized as 'other'. S: symbiotic; Ap: aposymbiotic and Ax: axenic.

Bacteroidetes) were represented with only few reads in aposymbiotic and axenic flies, respectively, but the Actinobacteria and Acidobacteria were markedly increased in relative terms.

Enterobacteriaceae (32.32 ± 5.07%) and Pseudomonaceae (2.13 ± 1.58%) (Proteobacteria) were the most dominant bacterial families detected in symbiotic flies, followed by Enterococcaceae (40.80 ± 4.98%) and Streptococcaceae (4.41 ± 1.78%) (Firmicutes) and Flavobacteriaceae (17.24 ± 5.07%) (Bacteroidetes). Bacteria from Colwelliaceae (Proteobacteria) family were not detected in aposymbiotic and axenic flies, whereas Corynebacterium (Actinobacteria) and Cyanobacteria proliferated significantly in aposymbiotic and axenic flies, respectively (for Corynebacterium: ANOVA, d.f. = 8; $F = 65.8689$; $p < 0.0001$ and d.f. = 8; $F = 65.8689$; $p < 0.0001$, respectively, and for Unidentified_chloroplast: ANOVA, d.f. = 8; $F = 126.0689$; $p < 0.0001$ and d.f. = 8; $F = 126.0689$; $p < 0.0001$, respectively) (figure 4).

*Myroides* (Flavobacteria), *Serratia, Colwellia* and *Pseudomonas* (Gammaproteobacteria) were the most abundant genera detected in symbiotic flies (with overall relative abundance of greater than or equal to 51.62%), while *Lactococcus* and *Vagococcus* (Bacilli) (with overall relative abundance of ≥19.25%), *Myroides*

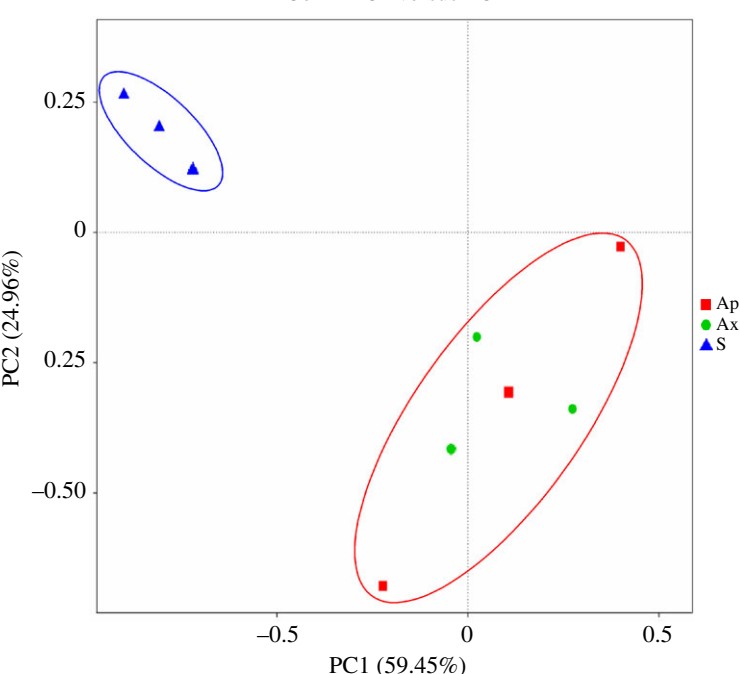

**Figure 4.** PCoA showing compositional differences between the bacterial community of *B. dorsalis* gut samples. Each point represents a biological replicate and different point colours represent different treatments (S: symbiotic; Ap: aposymbiotic and Ax: axenic). The distance between points represents differences in the microbial community composition between samples and between treatments. The circles show clusters of different bacterial samples.

and *Empedobacter* (Flavobacteria), *Corynebacterium* (Actinobacteria) and Unidentified_chloroplast (Cyanobacteria) were the only abundant genera detected in aposymbiotic and axenic flies, respectively (electronic supplementary material, figure S5).

On average, 133 and 345 OTUs disappeared from aposymbiotic and axenic flies, respectively, compared with the symbiotic ones. Only 47 OTUs were shared among replicates of all treatment groups (symbiotic, aposymbiotic and axenic) (electronic supplementary material, figure S2) and their identity is presented in electronic supplementary material, table S1. Among genera that disappeared from the communities of antibiotic and dechorionization treatments, we have *Chryseobacterium*, *Rhizobium*, *Dysgomonas*, *Cetobacterium*, *Sphingomonas*, *Colwellia*, *Prevotella*, and many unidentified species from Proteobacteria, Firmicutes, Bacteroidetes and Thaumarcheota (electronic supplementary material, S2).

## 3.5. Effect of symbiotic status, sex and diet quality on the longevity of *Bactrocera dorsalis*

Adult survival was significantly affected by the symbiotic status of flies, diet types and sex (Cox's regression model, HR = 1.30, $p < 0.0001$, HR = 1.13; $p < 0.0001$, and HR = 1.47; $p < 0.0001$, respectively). The mean longevity of flies under the experimental diets varied between 46 and 50 days for males (figure 5$a$,$c$) and between 35 to 61 days for females (figure 5$b$,$d$). Symbiotic flies recorded the highest lifespan regardless of sex and diet types in comparison with aposymbiotic and axenic flies, respectively (ANOVA, d.f. = 5; $F = 905.48$; $p < 0.0001$ and d.f. = 5; $F = 1381.491$; $p < 0.0001$, respectively). The longevity was sexually dimorphic as female flies lived longer when fed with full diet, and both female and male flies lived longer when fed with a full diet than when fed with a non-essential amino acids diet (ANOVA, d.f. = 5; $F = 1605.367$; $p = 0.0028$ and d.f. = 5; $F = 607.467$; $p < 0.0001$, respectively) (figure 5).

## 4. Discussion

Our results show the predominance of the phyla Proteobacteria, Firmicutes and Bacteroidetes (figure 3) and the families Enterobacteriaceae (75.96 ± 6.72%), Enterococcaceae (44.80 ± 4.98%), Streptococcaceae (94.31 ± 2.34%) and Flavobacteriaceae (47.32 ± 5.07%) in untreated flies (symbiotic control) (electronic

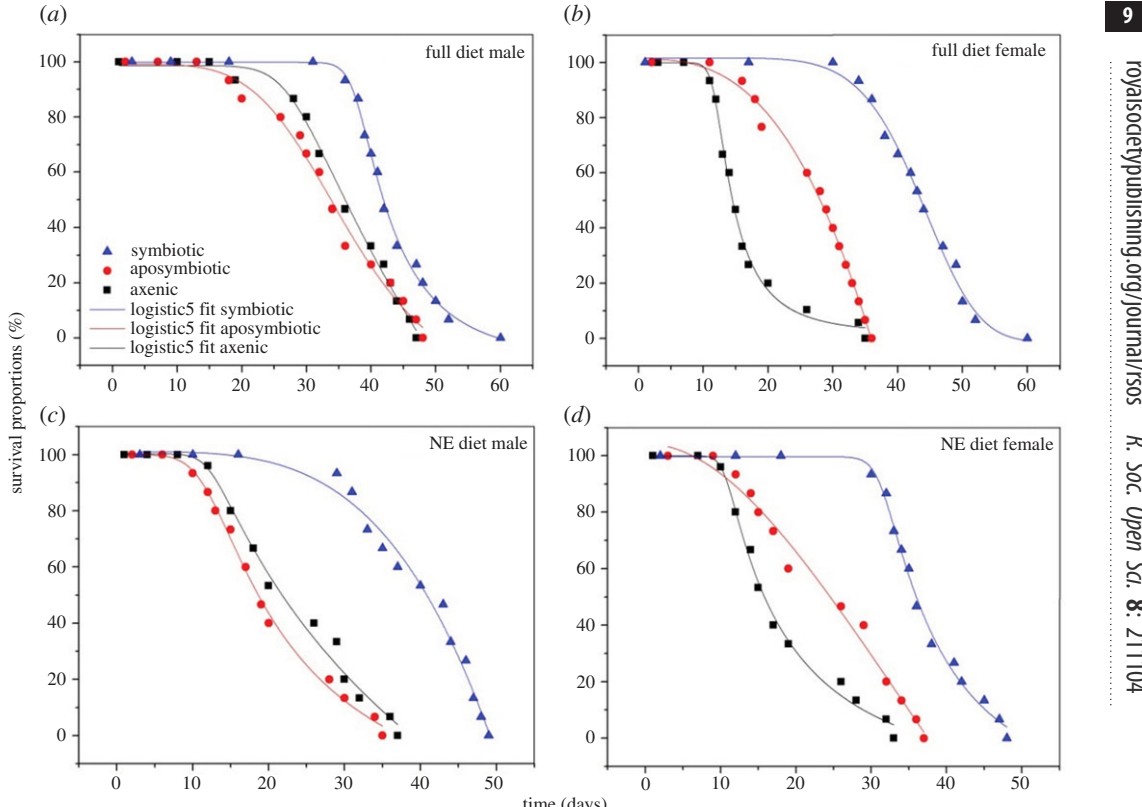

**Figure 5.** Survival curves symbiotic (blue triangle), aposymbiotic (red circle) and axenic (black cube) of males ((a,c) $p = 0.0028$) and females ((b,d) $p < 0.0001$) adult flies fed a full amino acids diet (a,b) and a non-essential amino acids diet (c,d).

supplementary material, figure S3). The antibiotic and the egg disinfection treatments significantly reduced the composition and abundance of gut microbiota as detected by qPCR and pyrosequencing analyses (figure 3; electronic supplementary material, figure S4). A similar trend was obtained when larvae and adult *B. dorsalis* were continuously fed a combination of three antibiotics (tetracycline, ampicillin and streptomycin), which resulted in the reduction of normally abundant bacteria [28]. The minor bacterial phyla (Actinobacteria, Cyanobacteria and Acidobacteria) from the symbiotic control persisted significantly in aposymbiotic and axenic flies, respectively (figure 3), similar to the families (Corynebacteriaceae and unidentified chloroplast) compared with their proportions in untreated flies. Adult longevity was affected by both the symbiotic status (longer lifespan in symbiotic flies) and diet quality (longer lifespan when fed with full diets), irrespective of sex (figure 5).

## 4.1. Effects of antibiotic treatment and eggs sterilization on gut bacterial community structure and composition

The present study reveals a larger number of OTUs belonging to more diverse clades from the microbiomes of symbiotic flies (by about fourfold) in comparison with what was previously reported in adult *B. dorsalis* [16,18,42,43]. This could be due to the pooling of five flies in each sample prior to sequencing, instead of sequencing individual fly per sample as done in the above-mentioned studies. The difference could also be due to different sequencing methods (technology and primers) or due to the origin (location and host plant) of the insect population. Proteobacteria, Firmicutes and Bacteroidetes were the most abundant bacterial phyla found in symbiotic flies, whereas Actinobacteria, Acidobacteria and five other phyla occurred at very low relative abundance. These results confirm previous reports in which Proteobacteria and Firmicutes were also dominantly represented in adult *B. dorsalis* [18,42,43]. Proteobacteria, Firmicutes and Bacteroidetes may be involved in protein metabolism and in energy production, host maintenance and survival, as previously reported [18]. Enterobacteriaceae, Enterococcaceae and Streptococcaceae were dominantly represented in the untreated flies, in compliance with previous studies in which Enterobacteriaceae and Enterococcaceae were also shown to be dominant bacterial families in adult *B. dorsalis* [16,18].

Overall, the antibiotics treatment and egg dechorionation process significantly reduced the microbial gut community size and altered its microbial composition and diversity. While many bacterial species disappeared following antibiotics treatments and dechorionization process (axenic) (e.g. *Chryseobacterium*, *Rhizobium*, *Dysgomonas*, *Cetobacterium*, *Sphingomonas*, *Colwellia*, *Prevotella*, etc.), some persisted and even increased their population size (e.g. *Vagococcus teuberi*, *Pseudomonas aeruginosa*, *Proteus mirabilis*, *Serratia marcescens*, *Providencia*, *Lactococcus lactis*, *Enterococcus durans*).

Interestingly, although reduced in abundance and diversity, the bacterial communities of the aposymbiotic and axenic flies did not differ quantitatively and qualitatively, and a similar residual or transient bacterium was observed, dominantly represented by *Pseudomonas*, *Providencia*, *Serratia marcescens* and *Lactococcus lactis* (electronic supplementary material, S2). These results suggest that some residual bacteria might have acquired resistance to Norfloxacin and Ceftazedime (administered antibiotics) and/or escalated dechorionation process to establish and expand its populations. This could be understood at two levels. Either the antibiotics Norfloxacin and Ceftazedime altered normally abundant bacterial species, and by allosteric competition allowed the minor ones to take over in size, or the residual bacteria highly expressed genes involved in resistance to both antibiotics used. The experimental insects were surface sterilized with the intention to reduce risks of environmental cross-contaminations. We therefore believe that bacterial species found after antibiotics treatments and egg dechorionation (residual) more likely originate from insect guts. In similar experimental setting, some bacterial species such as *Providencia*, *Pseudomonas* and *Serratia* were suggested to be highly associated with resistance to tetracycline [23,44]. In the same light, in the present study, *Providencia*, *Pseudomonas* and *Serratia* persisted and increased in size following Norfloxacin and Ceftazedime treatments as well as dechorionation process. This may reflect their involvement to tenacious resilience of host insect, either by natural resistance acquisition or by opportunistic establishment due to deleterious effects of antibiotics to susceptible bacteria [23].

Furthermore, *Pseudomonas* and *Serratia* were shown to be more prevalent in flies with antibiotics antecedent [23]. It is more obvious that the 7-day antibiotic treatments may have prompted Norfloxacin- and Ceftazedime-sensitive bacteria to increase in size and number to limit their detrimental effects, not only to bacteria themselves, but also to host insect. Evidently, the exact mechanism of innate or acquired resistance of the above-mentioned bacteria is open to further investigation, but at the very least, our results are consistent with previous studies in highlighting that antibiotic treatments and/or egg dechorionation may elicit resistant bacteria over time with the extent which varies with the type of antibiotics used and duration of treatment. The 47 core bacterial species found in this study (among which *Providencia*, *Pseudomonas* and *Serratia*) (electronic supplementary material, table S1) may putatively develop different levels of resistance to Norfloxacin and Ceftazedime and/or to egg dechorionation process.

In both treatments, a relative increase in Actinobacteria, Cyanobacteria and Acidobacteria was recorded. These populations may be resistant to the antibiotics/disinfection procedure used, either thanks to their shield location (biofilms or other physical protection) or a resistance acquired from the environment. In any case, their proliferation was restricted as the bacterial community sizes in the treated flies were much lower than the carrying capacity observed in the symbiotic control, suggesting constrains on their growth.

## 4.2. Effects of treatment, diet and sex on adult longevity

The results revealed that diet type, symbiotic status (symbiotic, aposymbiotic and axenic) and their interaction had a significant effect on the mean longevity of indiscriminately both sexes (figure 5). Adult longevity was higher when treated and symbiotic control flies were fed full diets compared with those fed non-essential amino acid diets (NE). Gut bacteria (i.e. in symbiotic flies) appear to significantly contribute to adult longevity. A recent study conducted on *Drosophila melanogaster* showed that gut microbial density (or quantity) could serve as a protein-rich source to enhance host protein nutrition and longevity [45]. In line with this, our result may suggest that, in addition to compensating missing nutrients in host diets, the oriental fruit fly could use the intestinal bacteria as an additional protein source to fuel their metabolic functions. Conversely, in the absence of gut bacteria (in aposymbiotic and axenic flies), longevity was reduced only when the flies were fed NE diets. This result suggests that the antibiotic and the dechorionization may have directly affected the flies. Finally, no significant difference in longevity was observed between aposymbiotic and axenic flies when fed a similar diet. The absence of difference in the longevity of aposymbiotic and axenic flies may come from the disappearance of bacteria beneficial to the host nutrition.

In many tephritid species, proteinacious diets are necessary for life expectancy and development of flies [1,14,15,46]. Moreover, it is hypothesized that host insect may acquire protein from the digestion

of own gut bacteria, and the gut bacteria could be involved in complex carbohydrates breakdown into readily available by-product for the host nutrition [24,47–49]. When the gut microbiome is intact, the provisioning of full diet (diet with full complement of amino acids) was shown to enhance the gut bacterial cultivation [14,25].

In our study, the antibiotics treatment and egg dechorionization reduced the fly longevity in a sex dimorphic scenario (females exhibited greater longevity than males). In the presence of gut bacteria (symbiotic treatment) and full diet, experimental flies (male and female) lived longer than those in which gut bacteria have been altered. These results suggest that some gut bacterial species may sustain the life expectancy of the host by either optimizing host nutrients intake from diets or by provisioning marginal amino acids to the host. Previously, *Pseudomonas, Proteus, Stenotrophomonas, Enterococcus faecalis* and *Klebsiella oxytoca* were implicated to improving host survivorship [31,46]. Some of these bacteria were detected in abundant proportion in our study and may be associated with the longevity of symbiotic flies observed. The decrease of longevity can either be due to the reduction of beneficial bacteria, or the development of resistant, yet deleterious bacteria, or to the negative effects of the antibiotics themselves (effects that could be observed after two generations in adult flies, in contrast with the days of antibiotics-free diet administration in our study) [23] and dechorionization treatments.

## 5. Conclusion

In this study, we evaluated the effect of antibiotic and egg disinfection on the structure and composition of residual gut bacterial communities and longevity of *Bactrocera dorsalis*. These procedures resulted in significant reduction of the abundant gut bacterial populations in untreated flies (Proteobacteria, Firmicutes and Bacteroidetes), while the minor ones (Actinobacteria, Cyanbacteria and Acidobacteria) persisted. These bacteria were either resistant to the antibiotics/disinfection procedure or they were protected by their location (biofilms or other physical protection) or they may have resulted from diet/host contaminations. Moreover, the treatments affected the adult longevity in a way which varied with the quality of diet consumed. The decrease of longevity can either be due to the reduction of bacterial diversity or to the negative effects of the antibiotics and dechorionization treatments to the host fly. Although further studies are needed to elucidate functional implications of bacterial community shift following antibiotics and dechorionization treatments, we can at least presume that the presence of bacteria was a determinant predictor of the fly longevity. As concrete perspective, it would be necessary to carry out metatranscriptomics analysis using the same experimental setting to unveil the functional effects of antibiotics and dechorionization treatments on gut bacteria and on host fitness parameters.

Ethics. This study is carried out in the laboratory of Insect Physiology and Biochemistry of the College of Plant Science and Technology, Huazhong Agricultural University (Wuhan, People's Republic of China). The fruit flies *Bactrocera dorsalis* are not endangered or protected species. Therefore, no permit was required for their collection and manipulation.

Data accessibility. The datasets supporting this article have been uploaded as part of the electronic supplementary material and deposited at Dryad accessible through the following link: https://doi.org/10.5061/dryad.xsj3tx9ds.

Authors' contributions. C.-Y.N. conceived research. M.A. designed the protocol, conducted the experiments and wrote the first draft of the manuscript. X.R., Y.W. and X.Q. analysed data. O.T., K.M.M.L. and R.A.N.N. edited and critically revised the manuscript. All authors read and approved this submission.

Competing interests. The authors declare no competing interests (financial or non-financial).

Funding. This study was funded by the National Natural Science Foundation of China (grant nos. 31972270 and 31661143045), International Atomic Energy Agency (grant nos. CRP No. 17153 and No. 18269), Agricultural public welfare industry research supported by Ministry of Agriculture of People's Republic of China (grant no. 201503137) and the Fundamental Research Funds for the Central Universities (grant no. 2662015PY148).

Acknowledgements. The authors are grateful to Professor Edouard Jurkevitch (Departments of Plant Pathology & Microbiology, Faculty of Agriculture, Food and Environment, Hebrew University of Jerusalem, POB 12, 76100 Rehovot, Israel) and two anonymous reviewers for their insightful comments on the early version of this manuscript.

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
