## [Peer Review File · Royal Society Open Science]

Review History

RSOS-210267.R0 (Original submission)

Review form: Reviewer 1

Is the manuscript scientifically sound in its present form?

No

Are the interpretations and conclusions justified by the results?

No

Is the language acceptable?

No

Do you have any ethical concerns with this paper?

No

Have you any concerns about statistical analyses in this paper?

Yes

Recommendation?

Reject

Comments to the Author(s)

Summary

The study aimed at assessing the effect of two methods, commonly used to remove symbionts (i.e. antibiotics and dechorionization) on fruit flies bacterial communities and at evaluating the longevity of the treated (i.e. aposymbiotic and axenic) and untreated (i.e. symbiotic) adult flies when exposed to different diets. The authors have found that both treatments have a similar effect: they decreased but not eliminated the bacterial diversity and changed community composition compared to the untreated flies. Symbiotic flies lived longer than the aposymbiotic and axenic ones, but flies of each treatment tended to live longer under full diet compared to diet with non-essential amino acids.

The strengths of the study is the comparison of two methods that remove insect symbiont, their effects on the insect bacterial communities (not often studied for this purpose) using a metabarcoding approach, and the link between the treatments and the diets since some bacteria play a role in their host nutrition by providing essential nutrients.

The limits of the study are that i) the data are underexploited; ii) the link between treatments and diet is under-developed; iii) the important results are not well-highlighted despite the study being an interesting subject, making the discussion a bit dull to read; iv) the way statistics are presented is confusing (see comments); v) the methods section is long and confusing because in the end I was still unsure of which and how many samples were used for which treatment, diet and experiment (see comments).

Major comments

1. Reorganizing the methods section, not necessarily shortening it, making a figure and explicitly indicating the number of replicates for each analysis would help to understand the method section more easily.
2. Figure 2 could be presented as supplemental materials as rarefaction curves are part of the data management prior to statistical analysis.
3. The table in supplemental data is unreadable. The authors should consider dispatching the different taxonomy levels in different columns. The meaning of each color used in the table should be indicated in the table caption. A heatmap would be more appropriate than a table.
4. The idea of core bacterial community (figure S1) is very interesting and would highlight the point about resistant bacteria. The authors should consider discussing this point and adding a table to present the taxonomy of the 47 OTUs. Is one of these OTUs dominant bacteria in term of abundance?
5. The dataset is rich but the discussion relies on the phyla that were detected. Proteobacteria is such a large phylum so hypothesizing about their role in the communities is a bit daring considering that species belonging to the same genus do not necessarily have the same function. The discussion would be more interesting if some results were discussed at the genus level though it is still hypothesis.
6. In the table in supplemental data, it seems that bacteria were identified at the species level, how is it possible when the authors used primers for 16S rRNA?
7. Also, the authors mentioned in the introduction that the fly microbiota is very rich and diverse but only a couple of genera was mentioned and only in supplemental data.
8. The authors should consider mentioning the number of OTUs and genera (and name) that disappeared from the communities of antibiotic and dechorionization-treated flies."
9. The authors should choose between the word "microbiome" and the word "microbiota" and use the same word throughout the whole manuscript. Same comment for "egg-sterilization" and "dechorionization".
10. The authors should revise the way they wrote the tests results (F value, df and P value, there is one value of each per factor in each ANOVA, there is 1 ANOVA per alpha diversity index

or the size of the community). P values from multiple comparisons could appear in figures and table as letters (commonly used in paper to indicate significant differences between modalities within one factor).

11. It is now frequent for authors of a paper on microbiota to make the raw sequences available on a platform (e.g. ENA), do the authors plan to do this?
12. The authors should consider discussing the bacteria resistant to antibiotics (they should be able to find papers on insect on this topic). Can bacteria “resist” dechorionization too?
13. The authors should also consider discussing the nutrition role of *B. dorsalis* bacteria (if some are known in the literature) in order to explain the longevity results and identifying if these bacteria were present in symbiotic flies and absent from antibiotic/dechorionized flies.

Minor comments

14. L60: what does “reduced the midgut microbiota” mean? Is it a decrease of diversity, of the abundance of certain bacteria or is it the number of bacteria that disappeared?
15. L61: there are more recent studies about the effects of antibiotics on insect microbiota than a study from 1972. See the following papers: Ourry et al 2020 (Long-lasting effects of antibiotics on bacterial communities of adult flies); Heys et al 2018 (The effect of gut microbiota elimination in *Drosophila melanogaster*: A how-to guide for host-microbiota studies); Lin et al 2015 (Evaluation of five antibiotics on larval gut bacterial diversity of *Plutella xylostella* (Lepidoptera: Plutellidae))
16. L71: in the sentence presenting the aim of the study, please add the Latin name of the studied insect to make it clear because several species were mentioned in the introduction
17. L96: please briefly describe the different diets composition (maybe in a table in supplemental data?). What is a full diet? It is also unclear which diets are fed to the flies in the different experiments/analyses.
18. L106: the fact that the insects were not fed for 2 days of antibiotics might not be enough as effects of antibiotics could be observed after two generations in adult flies (Ourry et al 2020). That could be something to discuss in the discussion, also in regard of antibiotic-resistant bacteria.
19. L114: the method section about the Validation of egg sterility could be a supplemental data and the authors could mention in the results section that egg sterility was checked and obtained.
20. L116: please mention the concerned regions of the 16 rRNA gene
21. L133: what is the point of measuring the gut bacterial community size with qPCR? Is it to obtain an absolute abundance in contrast to the relative abundance obtained after sequencing?
22. L152: what about the sugar diet? Why was the sugar diet excluded from the longevity assay? Did the authors make sure that a change in diet (from sugar to a full diet) did not decrease/increase the performance of symbiotic flies?
23. L210: Did the mortality data respect the normality?
24. L221: which distance matrix was used? Bray Curtis dissimilarity matrix is often used.
25. L225: is it really possible to obtain mean values for the beta diversity, especially when using a distance matrix?
26. L240: why did the authors use 5 indices for the alpha diversity? It is expected that all of them are discussed in the discussion. The most common ones are the number of observed OTUs (absent from the indices), Shannon and Simpson.
27. L240-250: in this paragraph, the F and P values are not correctly associated to each index. The way it is written, it could be interpreted as p value from multiple comparisons (Tukey test) between the 3 treatments and not ANOVA results. For each index ANOVA, there should be one F value, 1 df (with a value of 2 because there are 3 treatments) and 1 p value. P values to compare two treatments do not come from ANOVA but Tukey test.
28. The authors should consider adding letters in table 1 to indicate significant differences between symbiotic, aposymbiotic and axenic flies for each alpha diversity index.

29. L258: in the section title, do the authors mean CFU and not OTUs? Because OTUs are not mentioned in this section.
30. L260: how can CFU be obtained when a qPCR analysis was performed? Shouldn't it be a quantity of DNA instead of CFU?
31. L260-266: same comments on the test values. There should be 1 ANOVA and thus 1 F value, 1 df and 1 P value.
32. Figure S2: three (and not two) statistical results should be indicated in the figure, between symbiotic and aposymbiotic, between symbiotic and axenic, and between aposymbiotic and axenic.
33. L267: same comments on ANOVA. For bacterial relative abundance at different taxonomy levels, the authors should consider a Generalized Linear Model (poisson family and log link).
34. L304: usually the results about community structure (i.e. beta diversity) are presented after, or sometimes before, the alpha diversity results (i.e. richness and diversity indices), and not at the end of the results section
35. L305: what is the MRPP analysis? Is it a statistical analysis commonly performed on beta diversity? Again, why are there so many p values? What are the values for the test that determine whether the treatment factor has a significant effect or not on the beta diversity (i.e. community structure)?
36. L319: that is the correct way of writing the test result, but the result about the sex factor is missing.
37. L323-327: these test results should be indicated in the figure 5 as letters.
38. L335-337: is the reduction due to the egg sterilization consistent with the literature? Only the antibiotic treatment was discussed in the sentence.
39. L348: could the difference be due to different sequencing methods (technology, primers?) or due to the origin (location and host plant) of the insect population (i.e. where it was sampled in the field)?
40. L355: could it be possible to discuss bacterial functions at the genus level instead? Especially since Proteobacteria is a very large and diverse phylum.
41. L361-362: is it a result consistent with the literature?
42. L363-364: any explanation/hypothesis of why there was not difference between the antibiotics and egg sterilization treatments?
43. L365: can this result be discussed at the genus level? There are some papers that discussed about an increase of bacterial abundance after an antibiotic treatments and about bacteria being antibiotic-resistant. That could also be worth mentioning in this paper.
44. L380: are there any bacterial genera or species in *B. dorsalis* which are known to provide nutrients to their insect host and that were present in the bacterial communities of symbiotic flies and absent from the communities of antibiotic-treated and egg-sterilized flies of the present study? That could be interesting to discuss.
45. L384: "This result suggests that the antibiotic and the dechorionization may have directly affected the flies" What do the authors mean by directly? Are they refereeing to the toxicity of antibiotics and the negative effect of dechorionization or to the effect of each method on the insect microbiota (which correspond to an indirect effect in the latter case)? The antibiotic toxicity and negative effect of dechorionization should be discussed.
46. L385: The absence of difference in the longevity of aposymbiotic and axenic flies may come from the disappearance of bacteria beneficial to the host nutrition (see comments L380)
47. L387-388: How can "Actinobacteria, Cyanobacteria and Acidobacteria [...] have no potentials to extend fly longevity" when no bacteria from these phyla nor their functions were discussed priory?
48. L396: "horizontally transmitted", that is a new idea that was not discussed at all in the discussion. Do the authors mean contamination?

49. L400: the sentence is a bit too daring. The decrease of longevity can either be due to the reduction of bacterial diversity but also to the negative effects of the antibiotic and dechlorination treatments.
50. What could be a concrete perspective of this work?

Review form: Reviewer 2

Is the manuscript scientifically sound in its present form?

Yes

Are the interpretations and conclusions justified by the results?

Yes

Is the language acceptable?

Yes

Do you have any ethical concerns with this paper?

No

Have you any concerns about statistical analyses in this paper?

No

Recommendation?

Reject

Comments to the Author(s)

Many experimental studies on model organisms use axenic individuals or treatment with antibiotics to study the role of the gut microbiota on various host fitness parameters. If done appropriately, axenic individuals should, as per definition, be germ free. Antibiotic treatment, on the other hand, does not eliminate bacteria but reduces bacterial abundance and changes the microbial composition dramatically. I therefore think that experimental studies to compare axenic individuals with those treated by antibiotics potentially could be relevant and of interest to a general audience. Especially, if the magnitude and direction of changes to the gut microbial composition differs predictably between the two methods.

Unfortunately, this manuscript makes no effort to generalize the findings from the model species used. Especially the Discussion clearly does not place the findings of this study into the context of current knowledge in this field of research as very few references are included in the Discussion. Also, it is unclear what the implications of this study will be to the field. How do the findings affect future work on model organisms that aim to modify/eliminate gut bacteria? The study is therefore currently a very specific study on a specific model organism, which I believe misses the scope of this journal, and probably belongs in a more specialized journal with a narrower scope.

Other comments

In the Introduction, it should be made more clear what the purpose of the study is to the overall research topic. Why was this study conducted, and how does this model system related to other model systems?

Line 70-71: You obviously expect the treatments to differ, so maybe say here that the magnitude and direction of changes remain unknown in most cases, and that it is the objective of your study

to outline if the two ways of reducing gut bacteria produce predictable differences in bacterial composition.

Not necessary to summarize results in the Introduction. Instead, produce firm hypotheses/predictions that the experimental work was built upon.

Please provide an overview of the diets used. Although this is based on previously published work, it would be very helpful to see the diets briefly outlined here.

Line 396: "or they were horizontally transmitted to the host". This could be very important to the conclusions of your study. Yet it only appears towards the end of the conclusion. Did you describe bacteria from the diets and/or the environment?

Decision letter (RSOS-210267.R0)

Dear Dr Akami:

I write you in regards to manuscript RSOS-210267 entitled "Effects of symbiotic population impairment on microbiome composition and fly longevity of *Bactrocera dorsalis*" which you submitted to Royal Society Open Science.

While both reviewers found data of interest within the paper, they both raise a number of substantive concerns with the manuscript, including the premise of the paper, data interpretation and the context of the work with respect to developments in the wider field. In view of the comments of the reviewers and editors, found at the bottom of this letter, your manuscript has been rejected for publication.

Thank you for considering Royal Society Open Science for the publication of your research. I hope this decision will not discourage you from submitting manuscripts in the future.

Yours sincerely,

Royal Society Open Science Editorial Office
Royal Society Open Science
openscience@royalsociety.org

on behalf of Professor John Dalton (Associate Editor) and Steve Brown (Subject Editor)
openscience@royalsociety.org

Associate Editor Comments to Author (Professor John Dalton):

Associate Editor: 1

Comments to the Author:

The reviewers felt that the paper contained much valuable results; however these were not fully explored and taken advantage of. Both reviewers had difficulty with the premise of the paper, data interpretation and how it was presented in the broader field.

Associate Editor: 2
 Comments to the Author:
 (There are no comments.)

Reviewers' Comments to Author:

Reviewer: 1

Comments to the Author(s)

Summary

The study aimed at assessing the effect of two methods, commonly used to remove symbionts (i.e. antibiotics and dechorionization) on fruit flies bacterial communities and at evaluating the longevity of the treated (i.e. aposymbiotic and axenic) and untreated (i.e. symbiotic) adult flies when exposed to different diets. The authors have found that both treatments have a similar effect: they decreased but not eliminated the bacterial diversity and changed community composition compared to the untreated flies. Symbiotic flies lived longer than the aposymbiotic and axenic ones, but flies of each treatment tended to live longer under full diet compared to diet with non-essential amino acids.

The strengths of the study is the comparison of two methods that remove insect symbiont, their effects on the insect bacterial communities (not often studied for this purpose) using a metabarcoding approach, and the link between the treatments and the diets since some bacteria play a role in their host nutrition by providing essential nutrients.

The limits of the study are that i) the data are underexploited; ii) the link between treatments and diet is under-developed; iii) the important results are not well-highlighted despite the study being an interesting subject, making the discussion a bit dull to read; iv) the way statistics are presented is confusing (see comments); v) the methods section is long and confusing because in the end I was still unsure of which and how many samples were used for which treatment, diet and experiment (see comments).

Major comments

1. Reorganizing the methods section, not necessarily shortening it, making a figure and explicitly indicating the number of replicates for each analysis would help to understand the method section more easily.
2. Figure 2 could be presented as supplemental materials as rarefaction curves are part of the data management prior to statistical analysis.
3. The table in supplemental data is unreadable. The authors should consider dispatching the different taxonomy levels in different columns. The meaning of each color used in the table should be indicated in the table caption. A heatmap would be more appropriate than a table.
4. The idea of core bacterial community (figure S1) is very interesting and would highlight the point about resistant bacteria. The authors should consider discussing this point and adding a table to present the taxonomy of the 47 OTUs. Is one of these OTUs dominant bacteria in term of abundance?
5. The dataset is rich but the discussion relies on the phyla that were detected. Proteobacteria is such a large phylum so hypothesizing about their role in the communities is a bit daring considering that species belonging to the same genus do not necessarily have the same function. The discussion would be more interesting if some results were discussed at the genus level though it is still hypothesis.
6. In the table in supplemental data, it seems that bacteria were identified at the species level, how is it possible when the authors used primers for 16S rRNA?
7. Also, the authors mentioned in the introduction that the fly microbiota is very rich and diverse but only a couple of genera was mentioned and only in supplemental data.
8. The authors should consider mentioning the number of OTUs and genera (and name) that disappeared from the communities of antibiotic and dechorionization-treated flies."

9. The authors should choose between the word “microbiome” and the word “microbiota” and use the same word throughout the whole manuscript. Same comment for “egg-sterilization” and “dechorionization”.
10. The authors should revise the way they wrote the tests results (F value, df and P value, there is one value of each per factor in each ANOVA, there is 1 ANOVA per alpha diversity index or the size of the community). P values from multiple comparisons could appear in figures and table as letters (commonly used in paper to indicate significant differences between modalities within one factor).
11. It is now frequent for authors of a paper on microbiota to make the raw sequences available on a platform (e.g. ENA), do the authors plan to do this?
12. The authors should consider discussing the bacteria resistant to antibiotics (they should be able to find papers on insect on this topic). Can bacteria “resist” dechorionization too?
13. The authors should also consider discussing the nutrition role of *B. dorsalis* bacteria (if some are known in the literature) in order to explain the longevity results and identifying if these bacteria were present in symbiotic flies and absent from antibiotic/dechorionized flies.

Minor comments

14. L60: what does “reduced the midgut microbiota” mean? Is it a decrease of diversity, of the abundance of certain bacteria or is it the number of bacteria that disappeared?
15. L61: there are more recent studies about the effects of antibiotics on insect microbiota than a study from 1972. See the following papers: Ourry et al 2020 (Long-lasting effects of antibiotics on bacterial communities of adult flies); Heys et al 2018 (The effect of gut microbiota elimination in *Drosophila melanogaster*: A how-to guide for host-microbiota studies); Lin et al 2015 (Evaluation of five antibiotics on larval gut bacterial diversity of *Plutella xylostella* (Lepidoptera: Plutellidae))
16. L71: in the sentence presenting the aim of the study, please add the Latin name of the studied insect to make it clear because several species were mentioned in the introduction
17. L96: please briefly describe the different diets composition (maybe in a table in supplemental data?). What is a full diet? It is also unclear which diets are fed to the flies in the different experiments/analyses.
18. L106: the fact that the insects were not fed for 2 days of antibiotics might not be enough as effects of antibiotics could be observed after two generations in adult flies (Ourry et al 2020). That could be something to discuss in the discussion, also in regard of antibiotic-resistant bacteria.
19. L114: the method section about the Validation of egg sterility could be a supplemental data and the authors could mention in the results section that egg sterility was checked and obtained.
20. L116: please mention the concerned regions of the 16 rRNA gene
21. L133: what is the point of measuring the gut bacterial community size with qPCR? Is it to obtain an absolute abundance in contrast to the relative abundance obtained after sequencing?
22. L152: what about the sugar diet? Why was the sugar diet excluded from the longevity assay? Did the authors make sure that a change in diet (from sugar to a full diet) did not decrease/increase the performance of symbiotic flies?
23. L210: Did the mortality data respect the normality?
24. L221: which distance matrix was used? Bray Curtis dissimilarity matrix is often used.
25. L225: is it really possible to obtain mean values for the beta diversity, especially when using a distance matrix?
26. L240: why did the authors use 5 indices for the alpha diversity? It is expected that all of them are discussed in the discussion. The most common ones are the number of observed OTUs (absent from the indices), Shannon and Simpson.
27. L240-250: in this paragraph, the F and P values are not correctly associated to each index. The way it is written, it could be interpreted as p value from multiple comparisons (Tukey test) between the 3 treatments and not ANOVA results. For each index ANOVA, there should be one F value, 1 df (with a value of 2 because there are 3 treatments) and 1 p value. P values to compare two treatments do not come from ANOVA but Tukey test.

28. The authors should consider adding letters in table 1 to indicate significant differences between symbiotic, aposymbiotic and axenic flies for each alpha diversity index.
29. L258: in the section title, do the authors mean CFU and not OTUs? Because OTUs are not mentioned in this section.
30. L260: how can CFU be obtained when a qPCR analysis was performed? Shouldn't it be a quantity of DNA instead of CFU?
31. L260-266: same comments on the test values. There should be 1 ANOVA and thus 1 F value, 1 df and 1 P value.
32. Figure S2: three (and not two) statistical results should be indicated in the figure, between symbiotic and aposymbiotic, between symbiotic and axenic, and between aposymbiotic and axenic.
33. L267: same comments on ANOVA. For bacterial relative abundance at different taxonomy levels, the authors should consider a Generalized Linear Model (poisson family and log link).
34. L304: usually the results about community structure (i.e. beta diversity) are presented after, or sometimes before, the alpha diversity results (i.e. richness and diversity indices), and not at the end of the results section
35. L305: what is the MRPP analysis? Is it a statistical analysis commonly performed on beta diversity? Again, why are there so many p values? What are the values for the test that determine whether the treatment factor has a significant effect or not on the beta diversity (i.e. community structure)?
36. L319: that is the correct way of writing the test result, but the result about the sex factor is missing.
37. L323-327: these test results should be indicated in the figure 5 as letters.
38. L335-337: is the reduction due to the egg sterilization consistent with the literature? Only the antibiotic treatment was discussed in the sentence.
39. L348: could the difference be due to different sequencing methods (technology, primers?) or due to the origin (location and host plant) of the insect population (i.e. where it was sampled in the field)?
40. L355: could it be possible to discuss bacterial functions at the genus level instead? Especially since Proteobacteria is a very large and diverse phylum.
41. L361-362: is it a result consistent with the literature?
42. L363-364: any explanation/hypothesis of why there was not difference between the antibiotics and egg sterilization treatments?
43. L365: can this result be discussed at the genus level? There are some papers that discussed about an increase of bacterial abundance after an antibiotic treatments and about bacteria being antibiotic-resistant. That could also be worth mentioning in this paper.
44. L380: are there any bacterial genera or species in *B. dorsalis* which are known to provide nutrients to their insect host and that were present in the bacterial communities of symbiotic flies and absent from the communities of antibiotic-treated and egg-sterilized flies of the present study? That could be interesting to discuss.
45. L384: "This result suggests that the antibiotic and the dechorionization may have directly affected the flies" What do the authors mean by directly? Are they refereeing to the toxicity of antibiotics and the negative effect of dechorionization or to the effect of each method on the insect microbiota (which correspond to an indirect effect in the latter case)? The antibiotic toxicity and negative effect of dechorionization should be discussed.
46. L385: The absence of difference in the longevity of aposymbiotic and axenic flies may come from the disappearance of bacteria beneficial to the host nutrition (see comments L380)
47. L387-388: How can "Actinobacteria, Cyanobacteria and Acidobacteria [...] have no potentials to extend fly longevity" when no bacteria from these phyla nor their functions were discussed priory?
48. L396: "horizontally transmitted", that is a new idea that was not discussed at all in the discussion. Do the authors mean contamination?

49. L400: the sentence is a bit too daring. The decrease of longevity can either be due to the reduction of bacterial diversity but also to the negative effects of the antibiotic and dechlorination treatments.

50. What could be a concrete perspective of this work?

Reviewer: 2

Comments to the Author(s)

Many experimental studies on model organisms use axenic individuals or treatment with antibiotics to study the role of the gut microbiota on various host fitness parameters. If done appropriately, axenic individuals should, as per definition, be germ free. Antibiotic treatment, on the other hand, does not eliminate bacteria but reduces bacterial abundance and changes the microbial composition dramatically. I therefore think that experimental studies to compare axenic individuals with those treated by antibiotics potentially could be relevant and of interest to a general audience. Especially, if the magnitude and direction of changes to the gut microbial composition differs predictably between the two methods.

Unfortunately, this manuscript makes no effort to generalize the findings from the model species used. Especially the Discussion clearly does not place the findings of this study into the context of current knowledge in this field of research as very few references are included in the Discussion. Also, it is unclear what the implications of this study will be to the field. How do the findings affect future work on model organisms that aim to modify/eliminate gut bacteria? The study is therefore currently a very specific study on a specific model organism, which I believe misses the scope of this journal, and probably belongs in a more specialized journal with a narrower scope.

Other comments

In the Introduction, it should be made more clear what the purpose of the study is to the overall research topic. Why was this study conducted, and how does this model system related to other model systems?

Line 70-71: You obviously expect the treatments to differ, so maybe say here that the magnitude and direction of changes remain unknown in most cases, and that it is the objective of your study to outline if the two ways of reducing gut bacteria produce predictable differences in bacterial composition.

Not necessary to summarize results in the Introduction. Instead, produce firm hypotheses/predictions that the experimental work was built upon.

Please provide an overview of the diets used. Although this is based on previously published work, it would be very helpful to see the diets briefly outlined here.

Line 396: "or they were horizontally transmitted to the host". This could be very important to the conclusions of your study. Yet it only appears towards the end of the conclusion. Did you describe bacteria from the diets and/or the environment?

Author's Response to Decision Letter for (RSOS-210267.R0)

See Appendix A.

RSOS-211104.R0

Review form: Reviewer 3

Is the manuscript scientifically sound in its present form?

Yes

Are the interpretations and conclusions justified by the results?

Yes

Is the language acceptable?

Yes

Do you have any ethical concerns with this paper?

No

Have you any concerns about statistical analyses in this paper?

No

Recommendation?

Accept as is

Comments to the Author(s)

I find that all the major and minor comments raised in the framework of the previous revision have been adequately addressed.

There are still a couple of typos that could be easily fixed:

- in the conclusions the sentence at line 509: "These procedures resulted in significant reduction of the abundant gut bacterial populations in untreated flies" doesn't seem right as referring to the treated flies and not to the untreated.

- in the methods, line 123: "Symbiotic and aposymbiotic flies were produced from the wild larvae as mentioned above", should rather be "as mentioned below"

Decision letter (RSOS-211104.R0)

Dear Dr Akami,

I am pleased to inform you that your manuscript entitled "Effects of symbiotic population impairment on microbiome composition and longevity of *Bactrocera dorsalis*" is now accepted for publication in Royal Society Open Science.

Please ensure that you send to the editorial office an editable version of your accepted manuscript, and individual files for each figure and table included in your manuscript. You can send these in a zip folder if more convenient. Failure to provide these files may delay the

processing of your proof. You may disregard this request if you have already provided these files to the editorial office.

on behalf of Professor John Dalton (Associate Editor) and Steve Brown (Subject Editor)
openscience@royalsociety.org

Associate Editor Comments to Author (Professor John Dalton):

Associate Editor

Comments to the Author:

Thanks for making such a comprehensive responses to our reviewers.

Reviewer comments to Author:

Reviewer: 3

Comments to the Author(s)

I find that all the major and minor comments raised in the framework of the previous revision have been adequately addressed.

There are still a couple of typos that could be easily fixed:

- in the conclusions the sentence at line 509: "These procedures resulted in significant reduction of the abundant gut bacterial populations in untreated flies" doesn't seem right as referring to the treated flies and not to the untreated.

- in the methods, line 123: "Symbiotic and aposymbiotic flies were produced from the wild larvae as mentioned above", should rather be "as mentioned below"

Appendix A

Editorial Board

June 05, 2021

Dear Editor,

Please find attached a revised copy of the manuscript “**Effects of symbiotic population impairment on microbiome composition and fly longevity of *Bactrocera dorsalis***” by Akami *et al.*

We thank you and the 2 reviewers for critically going through the manuscript and raising all aspects of the work which require further amendments. Your valuable comments and suggestions have greatly helped to improve on the quality of our results. All the comments have been addressed according to reviewers' requirements and suggestions, unless otherwise explained and justified. Also, we would like to put to your notice that pairwise analyse (Pearson Chi-square) was applied when comparing aposymbiotic and axenic data and new references have been included where necessary.

We attached four items for this appeal, namely, a rebuttal letter containing a point by point address to reviewer comments, a marked-up and clean copies of the manuscript as well as a supporting file containing supplemental results from our experiments.

The page and line numbers from the replies to reviewer comments (from below pages) are referred to the marked-up copy. Thank you for considering this revision and we look forward to your final decision at your earliest convenience.

On behalf of all authors and with my best regards./.

Mazarin Akami, MD

➤ *Department of Biochemistry, Faculty of Science, University of Douala, P.O Box 24157 Douala, Cameroon.

➤ College of Plant Science & Technology, Huazhong Agricultural University, Wuhan 430070, China;

*Current address: Phone: (00237) 655 856 213

E-mail: makami1987@gmail.com

RSOS-210267 - (Effects of symbiotic population impairment on microbiome composition and fly longevity of *Bactrocera dorsalis*)

Reviewers' Comments to Author:

Reviewer: 1 (Comments to the Author(s))

Point by point responses

Major comments

1. Reorganizing the methods section, not necessarily shortening it, making a figure and explicitly indicating the number of replicates for each analysis would help to understand the method section more easily.

Reply: Done. A flowchart of the experimental design has been added to the manuscript. **Please see Figure 1, Page 30.**

2. Figure 2 could be presented as supplemental materials as rarefaction curves are part of the data management prior to statistical analysis.

Reply: Done. **Please see Figure S1, supplementary (ESM_1).** Thank you.

3. The table in supplemental data is unreadable. The authors should consider dispatching the different taxonomy levels in different columns. The meaning of each color used in the table should be indicated in the table caption. A heatmap would be more appropriate than a table.

Reply: Done. A heatmap has been added as supplementary, **please see Fig S1 (ESM_1).**

4. The idea of core bacterial community (figure S1) is very interesting and would highlight the point about resistant bacteria. The authors should consider discussing this point and adding a table to present the taxonomy of the 47 OTUs. Is one of these OTUs dominant bacteria in term of abundance?

Reply: Done. **Please see Table S1 (ESM_1).** In addition, the point about resistant bacteria has been discussed. **Please see Pages 19-21, lines 417-456.**

5. The dataset is rich but the discussion relies on the phyla that were detected. Proteobacteria is such a large phylum so hypothesizing about their role in the communities is a bit daring considering that species belonging to the same genus do not necessarily have the same function. The discussion would be more interesting if some results were discussed at the genus level though it is still hypothesis.

Reply: Done. **Please see Pages 19-21, lines 417-456.**

6. In the table in supplemental data, it seems that bacteria were identified at the species level, how is it possible when the authors used primers for 16S rRNA?

Reply: Thank you for this question. In fact, the sequencing of V1-V3 variable regions of the bacterial 16S rRNA gene could allow differentiation of bacterial community at species level by using broadly conserved primers (27F/533R) as indicated in the MM section (**Please see Pages 9-10, lines 190-195**).

7. Also, the authors mentioned in the introduction that the fly microbiota is very rich and diverse but only a couple of genera was mentioned and only in supplemental data.

Reply: This has been corrected. **Please see Table S1 of the supplementary ESM_1.**

8. The authors should consider mentioning the number of OTUs and genera (and name) that disappeared from the communities of antibiotic and dechorionization-treated flies.”

Reply: Done. **Please see Pages 16, lines 346-353.**

9. The authors should choose between the word “microbiome” and the word “microbiota” and use the same word throughout the whole manuscript. Same comment for “egg-sterilization” and “dechorionization”.

Reply: This has been corrected throughout the manuscript. Thank you.

10. The authors should revise the way they wrote the tests results (F value, df and P value, there is one value of each per factor in each ANOVA, there is 1 ANOVA per alpha diversity index or the size of the community). P values from multiple comparisons could appear in figures and table as letters (commonly used in paper to indicate significant differences between modalities within one factor).

Reply: Corrected. In addition, Pearson Chi Square was used for paired analysis between aposymbiotic and axenic data. **Please see Page 13, lines 272-277.**

11. It is now frequent for authors of a paper on microbiota to make the raw sequences available on a platform (e.g. ENA), do the authors plan to do this?

Reply: Yes, this manuscript is the first of a series at the end of which, fastq sequences will be deposited to SRA of NCBI. All the same, raw sequences can be made available to readers upon request.

12. The authors should consider discussing the bacteria resistant to antibiotics (they should be able to find papers on insect on this topic). Can bacteria “resist” dechorionization too?

Reply: Done. **Please see Pages 19-21, lines 417-456.**

13. The authors should also consider discussing the nutrition role of *B. dorsalis* bacteria (if some are known in the literature) in order to explain the longevity results and identifying if these bacteria were present in symbiotic flies and absent from antibiotic/dechorionized flies.

Reply: Done. **Please see Pages 22-23, lines 479-504.**

Minor comments

14. L60: what does “reduced the midgut microbiota” mean? Is it a decrease of diversity, of the abundance of certain bacteria or is it the number of bacteria that disappeared?

Reply: It is both, the decrease of community diversity and abundance. Precision has been made. **Please see Page 3, lines 61-62.**

15. L61: there are more recent studies about the effects of antibiotics on insect microbiota than a study from 1972. See the following papers: Ourry et al 2020 (Long-lasting effects of antibiotics on bacterial communities of adult flies); Heys et al 2018 (The effect of gut microbiota elimination in *Drosophila melanogaster*: A how-to guide for host–microbiota studies); Lin et al 2015 (Evaluation of five antibiotics on larval gut bacterial diversity of *Plutella xylostella* (Lepidoptera: Plutellidae))

Reply: Done. **Please see Page 4, lines 65-67.**

16. L71: in the sentence presenting the aim of the study, please add the Latin name of the studied insect to make it clear because several species were mentioned in the introduction

Reply: Done. **Please see Page 4, line 83.**

17. L96: please briefly describe the different diets composition (maybe in a table in supplemental data?). What is a full diet? It is also unclear which diets are fed to the flies in the different experiments/analyses.

Reply: Done. **Please see Page 6, lines 116-121.**

18. L106: the fact that the insects were not fed for 2 days of antibiotics might not be enough as effects of antibiotics could be observed after two generations in adult flies (Ourry et al 2020). That could be something to discuss in the discussion, also in regard of antibiotic-resistant bacteria.

Reply: Done. **Please see Pages 19-21, lines 417-456.**

19. L114: the method section about the Validation of egg sterility could be a supplemental data and the authors could mention in the results section that egg sterility was checked and obtained.

Reply: Done. **Please see Page 7, line 138.**

20. L116: please mention the concerned regions of the 16 rRNA gene

Reply: Done. **Please see section Validation of egg sterility on Page 7, supplementary (ESM_1).**

21. L133: what is the point of measuring the gut bacterial community size with qPCR? Is it to obtain an absolute abundance in contrast to the relative abundance obtained after sequencing?

Reply: Yes, the rationale of measuring the gut bacterial community size with qPCR is to evaluate the absolute abundance of the bacterial communities in host fly in comparison to the relative abundance obtained after sequencing.

22. L152: what about the sugar diet? Why was the sugar diet excluded from the longevity assay? Did the authors make sure that a change in diet (from sugar to a full diet) did not decrease/increase the performance of symbiotic flies?

Reply: The sugar diet was used just to maintain the flies during the seven days of protein starvation period. This procedure aims at evaluating the extent to which gut bacteria may constitute a source of protein to the host insect and/or help in synthesizing missing nutrients from host diets.

23. L210: Did the mortality data respect the normality?

Reply: Yes, precision has been made. **Please see Page 11, line 236.**

24. L221: which distance matrix was used? Bray Curtis dissimilarity matrix is often used.

Reply: Bray Curtis dissimilarity matrix was used, precision has been made. **Please see Page 12, line 247.**

25. L225: is it really possible to obtain mean values for the beta diversity, especially when using a distance matrix?

Reply: Each treatment had three biological replicates, reason why we calculated mean values as it is presented in **Table 1, Page 34.**

26. L240: why did the authors use 5 indices for the alpha diversity? It is expected that all of them are discussed in the discussion. The most common ones are the number of observed OTUs (absent from the indices), Shannon and Simpson.

Reply: We used the five indices to infer the total number of different bacterial species (species richness), as well as the relative abundance of each species in a sample (species evenness), as affected by antibiotics and dechlorination treatments. The observed OTUs appeared as observed species in table 1, this has been corrected (**Table 1, Page 34**). These results gave orientations for the discussion in light with the density of bacterial species and their evenness.

27. L240-250: in this paragraph, the F and P values are not correctly associated to each index. The way it is written, it could be interpreted as p value from multiple comparisons (Tukey test) between the 3 treatments and not ANOVA results. For each index ANOVA, there should be one F value, 1 df (with a value of 2 because there are 3 treatments) and 1 p value. P values to compare two treatments do not come from ANOVA but Tukey test.

Reply: ANOVA test was used when comparing indices of the three samples, paired dataset was analyzed by Pearson Chi square. This has been corrected. **Please see Page 13, lines 272-277.** Thank you for pointing it out.

28. The authors should consider adding letters in table 1 to indicate significant differences between symbiotic, aposymbiotic and axenic flies for each alpha diversity index.

Reply: Done. **Please see corrected version of Table 1, on Page 34.**

29. L258: in the section title, do the authors mean CFU and not OTUs? Because OTUs are not mentioned in this section.

Reply: Right. The title has been corrected and the whole section has been moved to the supplementary. **Please Page 8,** under the title Absolute gut bacterial abundance by qPCR (**ESM_1**).

30. L260: how can CFU be obtained when a qPCR analysis was performed? Shouldn't it be a quantity of DNA instead of CFU?

Reply: Right, corrected. **Please see Figure S4 supplementary, Page 9 (ESM_1).**

31. L260-266: same comments on the test values. There should be 1 ANOVA and thus 1 F value, 1 df and 1 P value.

Reply: Corrected. Please see supplementary, **Page 8** under the title Absolute gut bacterial abundance by qPCR (**ESM_1**).

32. Figure S2: three (and not two) statistical results should be indicated in the figure, between symbiotic and aposymbiotic, between symbiotic and axenic, and between aposymbiotic and axenic.

Reply: Right, our choice of presenting just two results stems from the fact there was no statistics difference between aposymbiotic and axenic treatments ($P > 0.05$), which both were highly different from the symbiotic one with similar p values ($P < 0.001$). Distinction has been made on figure caption. **Please see Figure S4, Page 9, supplementary (ESM_1).**

33. L267: same comments on ANOVA. For bacterial relative abundance at different taxonomy levels, the authors should consider a Generalized Linear Model (poisson family and log link).

Reply: ANOVA Corrected. We used Chi square test for paired analyses (aposymbiotic and axenic). **Please see Page 15, line 320.**

34. L304: usually the results about community structure (i.e. beta diversity) are presented after, or sometimes before, the alpha diversity results (i.e. richness and diversity indices), and not at the end of the results section

Reply: Corrected. The entire section has been placed after the diversity result. **Please see Pages 13-14, lines 280-292.**

35. L305: what is the MRPP analysis? Is it a statistical analysis commonly performed on beta diversity? Again, why are there so many p values? What are the values for the test that determine whether the treatment factor has a significant effect or not on the beta diversity (i.e. community structure)?

Reply: Right. The Multi Response Permutation Procedure (MRPP) is statistical analysis applied on beta diversity to evaluate relatedness of bacterial species within and between samples. It is based on Bray-Curtis distance matrix.

36. L319: that is the correct way of writing the test result, but the result about the sex factor is missing.

Reply: Thank you. Sex factor has been considered in the analysis. **Please see Pages 17-18, lines 371, 376-377.**

37. L323-327: these test results should be indicated in the figure 5 as letters.

Reply: We didn't put the letters to avoid overloading the figure. But we included some distinctions in the figure caption. **Please see the caption of Figure 5 on Page 34, line 738.**

38. L335-337: is the reduction due to the egg sterilization consistent with the literature? Only the antibiotic treatment was discussed in the sentence.

Reply: Yes, additional information has been added to the discussion. **Please see Pages 19-21, lines 418-457.**

39. L348: could the difference be due to different sequencing methods (technology, primers?) or due to the origin (location and host plant) of the insect population (i.e. where it was sampled in the field)?

Reply: Pertinent assumption, this has been included in the discussion section. **Please see Page 19, lines 402-404. Thank you.**

40. L355: could it be possible to discuss bacterial functions at the genus level instead? Especially since Proteobacteria is a very large and diverse phylum.

Reply: Done. **Please see Pages 19-21, lines 418-457.**

41. L361-362: is it a result consistent with the literature?

Reply: Yes, the entire sentence has been rephrased. **Please see Pages 19-21, lines 418-457.**

42. L363-364: any explanation/hypothesis of why there was not difference between the antibiotics and egg sterilization treatments?

Reply: These flies may have lost similar beneficial bacteria following antibiotics and dechorionization treatments. The sentence has been rephrased. **Please see Pages 19-21, lines 418-457.**

43. L365: can this result be discussed at the genus level? There are some papers that discussed about an increase of bacterial abundance after an antibiotic treatments and about bacteria being antibiotic-resistant. That could also be worth mentioning in this paper.

Reply: Done. **Please see Pages 19-21, lines 418-457.**

44. L380: are there any bacterial genera or species in *B. dorsalis* which are known to provide nutrients to their insect host and that were present in the bacterial communities of symbiotic flies and absent from the communities of antibiotic-treated and egg-sterilized flies of the present study? That could be interesting to discuss.

Reply: Done. **Please see Pages 22-23, lines 479-504.**

45. L384: “This result suggests that the antibiotic and the dechorionization may have directly affected the flies” What do the authors mean by directly? Are they refereeing to the toxicity of antibiotics and the negative effect of dechorionization or to the effect of each method on the insect microbiota (which correspond to an indirect effect in the latter case)? The antibiotic toxicity and negative effect of dechorionization should be discussed.

Reply: Done. **Please see Pages 23, lines 500-504.**

46. L385: The absence of difference in the longevity of aposymbiotic and axenic flies may come from the disappearance of bacteria beneficial to the host nutrition (see comments L380).

Reply: Corrected. **Please see Page 22, lines 479-483.**

47. L387-388: How can “Actinobacteria, Cyanobacteria and Acidobacteria [...] have no potentials to extend fly longevity” when no bacteria from these phyla nor their functions were discussed priory?

Reply: Rephrased. **Please see Pages 23, lines 491-504.**

48. L396: “horizontally transmitted”, that is a new idea that was not discussed at all in the discussion. Do the authors mean contamination?

Reply: Yes, contamination is more appropriate to be used. This has been corrected. **Please see Page 24, lines 512-513, thank you.**

49. L400: the sentence is a bit too daring. The decrease of longevity can either be due to the reduction of bacterial diversity but also to the negative effects of the antibiotic and dechorionization treatments.

Reply: Corrected. **Please see Page 24, lines 515-521.**

50. What could be a concrete perspective of this work?

Reply: Carry out metatranscriptomics analysis using the same experimental setting to unveil the functional effects of antibiotics and dechorionization treatments on gut bacteria and host fly. **Please see Page 24, lines 521-524.**

Reviewer: 2 (Comments to the Author(s))

Many experimental studies on model organisms use axenic individuals or treatment with antibiotics to study the role of the gut microbiota on various host fitness parameters. If done appropriately, axenic individuals should, as per definition, be germ free. Antibiotic treatment, on the other hand, does not eliminate bacteria but reduces bacterial abundance and changes the microbial composition dramatically. I therefore think that experimental studies to compare axenic individuals with those treated by antibiotics potentially could be relevant and of interest to a general audience. Especially, if the magnitude and direction of changes to the gut microbial composition differs predictably between the two methods.

Reply: Thank you for this interesting and faithful highlight, worth to be used as flagship of this manuscript.

Unfortunately, this manuscript makes no effort to generalize the findings from the model species used. Especially the Discussion clearly does not place the findings of this study into the context of current knowledge in this field of research as very few references are included in the Discussion. Also, it is unclear what the implications of this study will be to the field. How do the findings affect future work on model organisms that aim to modify/eliminate gut bacteria? The study is therefore currently a very specific study on a specific model organism, which I believe misses the scope of this journal, and probably belongs in a more specialized journal with a narrower scope.

Reply: The study prospects have been broadened. Please see the revised versions of the introduction and Discussion sections.

Other comments

In the Introduction, it should be made more clear what the purpose of the study is to the overall research topic. Why was this study conducted, and how does this model system related to other model systems?

Reply: The introduction section has been revised. **Please see Pages 3-5, Lines 42-102.**

Line 70-71: You obviously expect the treatments to differ, so maybe say here that the magnitude and direction of changes remain unknown in most cases, and that it is the objective of your study to outline if the two ways of reducing gut bacteria produce predictable differences in bacterial composition.

Reply: Done. **Please see Page 4, Lines 74-80. Thank you.**

Not necessary to summarize results in the Introduction. Instead, produce firm hypotheses/predictions that the experimental work was built upon.

Reply: Done. **Please see Page 5, Lines 85-102.** Thank you for pointing this out.

Please provide an overview of the diets used. Although this is based on previously published work, it would be very helpful to see the diets briefly outlined here.

Reply: Done. **Please see Table S2, Page 7, supplementary (ESM_1).**

Line 396: "or they were horizontally transmitted to the host". This could be very important to the conclusions of your study. Yet it only appears towards the end of the conclusion. Did you describe bacteria from the diets and/or the environment?

Reply: We did not describe bacteria from the diets and/or the environment. The assumption has been corrected and changed to contaminations. **Please see Pages 24, Lines 512-513.**